# Tent: Fully Test-Time Adaptation by Entropy Minimization

**Dequan Wang**[1]*, **Evan Shelhamer**[2]*[†], **Shaoteng Liu**[1], **Bruno Olshausen**[1], **Trevor Darrell**[1]
dqwang@cs.berkeley.edu, shelhamer@google.com
UC Berkeley[1]    Adobe Research[2]

## Abstract

A model must adapt itself to generalize to new and different data during testing. In this setting of fully test-time adaptation the model has only the test data and its own parameters. We propose to adapt by test entropy minimization (tent[1]): we optimize the model for confidence as measured by the entropy of its predictions. Our method estimates normalization statistics and optimizes channel-wise affine transformations to update online on each batch. Tent reduces generalization error for image classification on corrupted ImageNet and CIFAR-10/100 and reaches a new state-of-the-art error on ImageNet-C. Tent handles source-free domain adaptation on digit recognition from SVHN to MNIST/MNIST-M/USPS, on semantic segmentation from GTA to Cityscapes, and on the VisDA-C benchmark. These results are achieved in one epoch of test-time optimization without altering training.

## 1 Introduction

Deep networks can achieve high accuracy on training and testing data from the same distribution, as evidenced by tremendous benchmark progress (Krizhevsky et al., 2012; Simonyan & Zisserman, 2015; He et al., 2016). However, generalization to new and different data is limited (Hendrycks & Dietterich, 2019; Recht et al., 2019; Geirhos et al., 2018). Accuracy suffers when the training (source) data differ from the testing (target) data, a condition known as *dataset shift* (Quionero-Candela et al., 2009). Models can be sensitive to shifts during testing that were not known during training, whether natural variations or corruptions, such as unexpected weather or sensor degradation. Nevertheless, it can be necessary to deploy a model on different data distributions, so adaptation is needed.

During testing, the model must adapt given only its parameters and the target data. This *fully test-time adaptation* setting cannot rely on source data or supervision. Neither is practical when the model first encounters new testing data, before it can be collected and annotated, as inference must go on. Real-world usage motivates fully test-time adaptation by data, computation, and task needs:

1. Availability. A model might be distributed without source data for bandwidth, privacy, or profit.
2. Efficiency. It might not be computationally practical to (re-)process source data during testing.
3. Accuracy. A model might be too inaccurate without adaptation to serve its purpose.

To adapt during testing we minimize the entropy of model predictions. We call this objective the test entropy and name our method *tent* after it. We choose entropy for its connections to error and shift. Entropy is related to error, as more confident predictions are all-in-all more correct (Figure 1). Entropy is related to shifts due to corruption, as more corruption results in more entropy, with a strong rank correlation to the loss for image classification as the level of corruption increases (Figure 2).

To minimize entropy, tent normalizes and transforms inference on target data by estimating statistics and optimizing affine parameters batch-by-batch. This choice of low-dimensional, channel-wise feature modulation is efficient to adapt during testing, even for online updates. Tent does not restrict or alter model training: it is independent of the source data given the model parameters. If the model can be run, it can be adapted. Most importantly, tent effectively reduces not just entropy but error.

---

*Equal contribution. [†]Work done at Adobe Research; the author is now at DeepMind.
[1]Please see the project page at https://github.com/DequanWang/tent for the code and more.

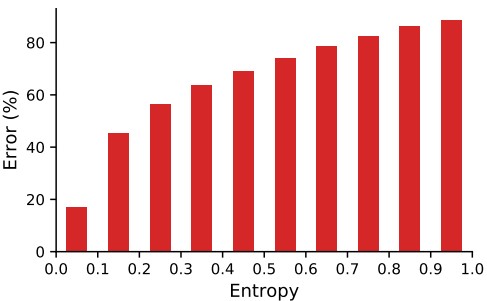
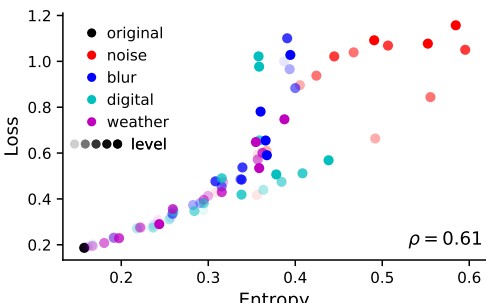

Figure 1: Predictions with lower entropy have lower error rates on corrupted CIFAR-100-C. Certainty can serve as supervision during testing.

Figure 2: More corruption causes more loss and entropy on CIFAR-100-C. Entropy can estimate the degree of shift without training data or labels.

Our results evaluate generalization to corruptions for image classification, to domain shift for digit recognition, and to simulation-to-real shift for semantic segmentation. For context with more data and optimization, we evaluate methods for robust training, domain adaptation, and self-supervised learning given the labeled source data. Tent can achieve less error given only the target data, and it improves on the state-of-the-art for the ImageNet-C benchmark. Analysis experiments support our entropy objective, check sensitivity to the amount of data and the choice of parameters for adaptation, and back the generality of tent across architectures.

**Our contributions**

- We highlight the setting of fully test-time adaptation with only target data and no source data. To emphasize practical adaptation during inference we benchmark with offline and online updates.

- We examine entropy as an adaptation objective and propose tent: a test-time entropy minimization scheme to reduce generalization error by reducing the entropy of model predictions on test data.

- For robustness to corruptions, tent reaches $44.0\%$ error on ImageNet-C, better than the state-of-the-art for robust training ($50.2\%$) and the strong baseline of test-time normalization ($49.9\%$).

- For domain adaptation, tent is capable of online and source-free adaptation for digit classification and semantic segmentation, and can even rival methods that use source data and more optimization.

## 2 SETTING: FULLY TEST-TIME ADAPTATION

Adaptation addresses generalization from source to target. A model $f_\theta(x)$ with parameters $\theta$ trained on source data and labels $x^s, y^s$ may not generalize when tested on shifted target data $x^t$. Table 1 summarizes adaptation settings, their required data, and types of losses. Our fully test-time adaptation setting uniquely requires only the model $f_\theta$ and unlabeled target data $x^t$ for adaptation during inference.

Existing adaptation settings extend training given more data and supervision. Transfer learning by fine-tuning (Donahue et al., 2014; Yosinski et al., 2014) needs target labels to (re-)train with a supervised loss $L(x^t, y^t)$. Without target labels, our setting denies this supervised training. Domain adaptation (DA) (Quionero-Candela et al., 2009; Saenko et al., 2010; Ganin & Lempitsky, 2015; Tzeng et al., 2015) needs both the source and target data to train with a cross-domain loss $L(x^s, x^t)$. Test-time training (TTT) (Sun et al., 2019b) adapts during testing but first alters training to jointly optimize its supervised loss $L(x^s, y^s)$ and self-supervised loss $L(x^s)$. Without source, our setting denies joint training across domains (DA) or losses (TTT). Existing settings have their purposes, but do not cover all practical cases when source, target, or supervision are not simultaneously available.

Unexpected target data during testing requires test-time adaptation. TTT and our setting adapt the model by optimizing an unsupervised loss during testing $L(x^t)$. During training, TTT jointly optimizes this same loss on source data $L(x^s)$ with a supervised loss $L(x^s, y^s)$, to ensure the parameters $\theta$ are shared across losses for compatibility with adaptation by $L(x^t)$. Fully test-time adaptation is independent of the training data and training loss given the parameters $\theta$. By not changing training, our setting has the potential to require less data and computation for adaptation.

Table 1: Adaptation settings differ by their data and therefore losses during training and testing. Of the source $^s$ and target $^t$ data $x$ and labels $y$, our fully test-time setting only needs the target data $x^t$.

| setting | source data | target data | train loss | test loss |
|---|---|---|---|---|
| fine-tuning | - | $x^t, y^t$ | $L(x^t, y^t)$ | - |
| domain adaptation | $x^s, y^s$ | $x^t$ | $L(x^s, y^s) + L(x^s, x^t)$ | - |
| test-time training | $x^s, y^s$ | $x^t$ | $L(x^s, y^s) + L(x^s)$ | $L(x^t)$ |
| fully test-time adaptation | - | $x^t$ | - | $L(x^t)$ |

(a) training      (b) fully test-time adaptation

Figure 3: Method overview. Tent does not alter training (a), but minimizes the entropy of predictions during testing (b) over a constrained modulation $\Delta$, given the parameters $\theta$ and target data $x^t$.

## 3   METHOD: TEST ENTROPY MINIMIZATION VIA FEATURE MODULATION

We optimize the model during testing to minimize the entropy of its predictions by modulating its features. We call our method *tent* for test entropy. Tent requires a compatible model, an objective to minimize (Section 3.1), and parameters to optimize over (Section 3.2) to fully define the algorithm (Section Section 3.3). Figure 3 outlines our method for fully test-time adaptation.

The model to be adapted must be trained for the supervised task, probabilistic, and differentiable. No supervision is provided during testing, so the model must already be trained. Measuring the entropy of predictions requires a distribution over predictions, so the model must be probabilistic. Gradients are required for fast iterative optimization, so the model must be differentiable. Typical deep networks for supervised learning satisfy these model requirements.

### 3.1   ENTROPY OBJECTIVE

Our test-time objective $L(x_t)$ is to minimize the entropy $H(\hat{y})$ of model predictions $\hat{y} = f_\theta(x^t)$. In particular, we measure the Shannon entropy (Shannon, 1948), $H(\hat{y}) = -\sum_c p(\hat{y}_c) \log p(\hat{y}_c)$ for the probability $\hat{y}_c$ of class $c$. Note that optimizing a single prediction has a trivial solution: assign all probability to the most probable class. We prevent this by jointly optimizing batched predictions over parameters that are shared across the batch.

Entropy is an unsupervised objective because it only depends on predictions and not annotations. However, as a measure of the predictions it is directly related to the supervised task and model.

In contrast, proxy tasks for self-supervised learning are not directly related to the supervised task. Proxy tasks derive a self-supervised label $y'$ from the input $x_t$ without the task label $y$. Examples of these proxies include rotation prediction (Gidaris et al., 2018), context prediction (Doersch et al., 2015), and cross-channel auto-encoding (Zhang et al., 2017). Too much progress on a proxy task could interfere with performance on the supervised task, and self-supervised adaptation methods have to limit or mix updates accordingly (Sun et al., 2019b;a). As such, care is needed to choose a proxy compatible with the domain and task, to design the architecture for the proxy model, and to balance optimization between the task and proxy objectives. Our entropy objective does not need such efforts.

### 3.2   MODULATION PARAMETERS

The model parameters $\theta$ are a natural choice for test-time optimization, and these are the choice of prior work for train-time entropy minimization (Grandvalet & Bengio, 2005; Dhillon et al., 2020; Carlucci et al., 2017). However, $\theta$ is the only representation of the training/source data in our setting, and altering $\theta$ could cause the model to diverge from its training. Furthermore, $f$ can be nonlinear and $\theta$ can be high dimensional, making optimization too sensitive and inefficient for test-time usage.

$$\text{normalization} \quad \mu \leftarrow \mathbb{E}[x_t], \sigma^2 \leftarrow \mathbb{E}[(\mu - x_t)^2]$$
$$\text{transformation} \quad \gamma \leftarrow \gamma + \partial H / \partial \gamma, \beta \leftarrow \beta + \partial H / \partial \beta$$

Figure 4: Tent modulates features during testing by estimating normalization statistics $\mu, \sigma$ and optimizing transformation parameters $\gamma, \beta$. Normalization and transformation apply channel-wise scales and shifts to the features. The statistics and parameters are updated on target data without use of source data. In practice, adapting $\gamma, \beta$ is efficient because they make up <1% of model parameters.

For stability and efficiency, we instead only update feature modulations that are linear (scales and shifts), and low-dimensional (channel-wise). Figure 4 shows the two steps of our modulations: normalization by statistics and transformation by parameters. Normalization centers and standardizes the input $x$ into $\bar{x} = (x - \mu)/\sigma$ by its mean $\mu$ and standard deviation $\sigma$. Transformation turns $\bar{x}$ into the output $x' = \gamma \bar{x} + \beta$ by affine parameters for scale $\gamma$ and shift $\beta$. Note that the statistics $\mu, \sigma$ are estimated from the data while the parameters $\gamma, \beta$ are optimized by the loss.

For implementation, we simply repurpose the normalization layers of the source model. We update their normalization statistics and affine parameters for all layers and channels during testing.

### 3.3 ALGORITHM

**Initialization** The optimizer collects the affine transformation parameters $\{\gamma_{l,k}, \beta_{l,k}\}$ for each normalization layer $l$ and channel $k$ in the source model. The remaining parameters $\theta \setminus \{\gamma_{l,k}, \beta_{l,k}\}$ are fixed. The normalization statistics $\{\mu_{l,k}, \sigma_{l,k}\}$ from the source data are discarded.

**Iteration** Each step updates the normalization statistics and transformation parameters on a batch of data. The normalization statistics are estimated for each layer in turn, during the forward pass. The transformation parameters $\gamma, \beta$ are updated by the gradient of the prediction entropy $\nabla H(\hat{y})$, during the backward pass. Note that the transformation update follows the prediction for the current batch, and so it only affects the next batch (unless forward is repeated). This needs just one gradient per point of additional computation, so we use this scheme by default for efficiency.

**Termination** For online adaptation, no termination is necessary, and iteration continues as long as there is test data. For offline adaptation, the model is first updated and then inference is repeated. Adaptation may of course continue by updating for multiple epochs.

## 4 EXPERIMENTS

We evaluate tent for corruption robustness on CIFAR-10/CIFAR-100 and ImageNet, and for domain adaptation on digit adaptation from SVHN to MNIST/MNIST-M/USPS. Our implementation is in PyTorch (Paszke et al., 2019) with the `pycls` library (Radosavovic et al., 2019).

**Datasets** We run on image classification datasets for corruption and domain adaptation conditions. For large-scale experiments we choose ImageNet (Russakovsky et al., 2015), with 1,000 classes, a training set of 1.2 million, and a validation set of 50,000. For experiments at an accessible scale we choose CIFAR-10/CIFAR-100 (Krizhevsky, 2009), with 10/100 classes, a training set of 50,000, and a test set of 10,000. For domain adaptation we choose SVHN (Netzer et al., 2011) as source and MNIST (LeCun et al., 1998)/MNIST-M (Ganin & Lempitsky, 2015)/USPS (Hull, 1994) as targets, with ten classes for the digits 0–9. SVHN has color images of house numbers from street views with a training set of 73,257 and test set of 26,032. MNIST/MNIST-M/USPS have handwritten digits with a training sets of 60,000/60,000/7,291 and test sets of 10,000/10,000/2,007.

**Models** For corruption we use residual networks (He et al., 2016) with 26 layers (R-26) on CIFAR-10/100 and 50 layers (R-50) on ImageNet. For domain adaptation we use the R-26 architecture. For fair comparison, all methods in each experimental condition share the same architecture.

Our networks are equipped with batch normalization (Ioffe & Szegedy, 2015). For the source model without adaptation, the normalization statistics are estimated during training on the source data. For all test-time adaptation methods, we estimate these statistics during testing on the target data, as done in concurrent work on adaptation by normalization (Schneider et al., 2020; Nado et al., 2020).

Table 2: Corruption benchmark on CIFAR-10-C and CIFAR-100-C for the highest severity. Tent has least error, with less optimization than domain adaptation (RG, UDA-SS) and test-time training (TTT), and improves on test-time norm (BN).

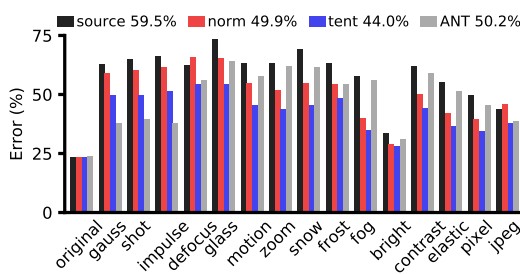

| Method | Source | Target | Error (%) C10-C | Error (%) C100-C |
|---|---|---|---|---|
| Source | train | | 40.8 | 67.2 |
| RG | train | train | 18.3 | 38.9 |
| UDA-SS | train | train | 16.7 | 47.0 |
| TTT | train | test | 17.5 | 45.0 |
| BN | | test | 17.3 | 42.6 |
| PL | | test | 15.7 | 41.2 |
| Tent (ours) | | test | **14.3** | **37.3** |

Figure 5: Corruption benchmark on ImageNet-C: error for each type averaged over severity levels. Tent improves on the prior state-of-the-art, adversarial noise training (Rusak et al., 2020), by fully test-time adaptation *without altering training*.

**Optimization** We optimize the modulation parameters $\gamma, \beta$ following the training hyperparameters for the source model with few changes. On ImageNet we optimize by SGD with momentum; on other datasets we optimize by Adam (Kingma & Ba, 2015). We lower the batch size (BS) to reduce memory usage for inference, then lower the learning rate (LR) by the same factor to compensate (Goyal et al., 2017). On ImageNet, we set BS = 64 and LR = 0.00025, and on other datasets we set BS = 128 and LR = 0.001. We control for ordering by shuffling and sharing the order across methods.

**Baselines** We compare to domain adaptation, self-supervision, normalization, and pseudo-labeling:

- source applies the trained classifier to the test data without adaptation,
- adversarial domain adaptation (RG) reverses the gradients of a domain classifier on source and target to optimize for a domain-invariant representation (Ganin & Lempitsky, 2015),
- self-supervised domain adaptation (UDA-SS) jointly trains self-supervised rotation and position tasks on source and target to optimize for a shared representation (Sun et al., 2019a),
- test-time training (TTT) jointly trains for supervised and self-supervised tasks on source, then keeps training the self-supervised task on target during testing (Sun et al., 2019b),
- test-time normalization (BN) updates batch normalization statistics (Ioffe & Szegedy, 2015) on the target data during testing (Schneider et al., 2020; Nado et al., 2020),
- pseudo-labeling (PL) tunes a confidence threshold, assigns predictions over the threshold as labels, and then optimizes the model to these pseudo-labels before testing (Lee, 2013).

Only test-time normalization (BN), pseudo-labeling (PL), and tent (ours) are fully test-time adaptation methods. See Section 2 for an explanation and contrast with domain adaptation and test-time training.

### 4.1 ROBUSTNESS TO CORRUPTIONS

To benchmark robustness to corruption, we make use of common image corruptions (see Appendix A for examples). The CIFAR-10/100 and ImageNet datasets are turned into the CIFAR-10/100-C and ImageNet-C corruption benchmarks by duplicating their test/validation sets and applying 15 types of corruptions at five severity levels (Hendrycks & Dietterich, 2019).

**Tent improves more with less data and computation.** Table 2 reports errors averaged over corruption types at the severest level of corruption. On CIFAR-10/100-C we compare all methods, including those that require joint training across domains or losses, given the convenient sizes of these datasets. Adaptation is offline for fair comparison with offline baselines. Tent improves on the fully test-time adaptation baselines (BN, PL) but also the domain adaptation (RG, UDA-SS) and test-time training (TTT) methods that need several epochs of optimization on source and target.

**Tent consistently improves across corruption types.** Figure 5 plots the error for each corruption type averaged over corruption levels on ImageNet-C. We compare the most efficient methods—source, normalization, and tent—given the large scale of the source data (>1 million images) needed by other methods and the 75 target combinations of corruption types and levels. Tent and BN adapt online to rival the efficiency of inference without adaptation. Tent reaches the least error for most corruption types without increasing the error on the original data.

Table 3: Digit domain adaptation from SVHN to MNIST/MNIST-M/USPS. Source-free adaptation is not only feasible, but more efficient. Tent always improves on normalization (BN), and in 2/3 cases achieves less error than domain adaptation (RG, UDA-SS) without joint training on source & target.

| Method | Source | Target | Epochs | Error (%) | | |
|--------|--------|--------|--------|-----------|-----------|------|
| | | | Source + Target | MNIST | MNIST-M | USPS |
| Source | train | | - | 18.2 | 39.7 | 19.3 |
| RG | train | train | 10 + 10 | 15.0 | 33.4 | 18.9 |
| UDA-SS | train | train | 10 + 10 | 11.1 | **22.2** | 18.4 |
| BN | | test | 0 + 1 | 15.7 | 39.7 | 18.0 |
| Tent (ours) | | test | 0 + 1 | 10.0 | 37.0 | 16.3 |
| Tent (ours) | | test | 0 + 10 | **8.2** | 36.8 | **14.4** |

**Tent reaches a new state-of-the-art without altering training.** The state-of-the-art methods for robustness extend training with adversarial noise (ANT) (Rusak et al., 2020) for $50.2\%$ error or mixtures of data augmentations (AugMix) (Hendrycks et al., 2020) for $51.7\%$ error. Combined with stylization from external images (SIN) (Geirhos et al., 2019), ANT+SIN reaches $47.4\%$. Tent reaches a new state-of-the-art of $44.0\%$ by online adaptation and $42.3\%$ by offline adaptation. It improves on ANT for all types except noise, on which ANT is trained. This requires just one gradient per test point, without more optimization on the training set (ANT, AugMix) or use of external images (SIN). Among fully test-time adaptation methods, tent reduces the error beyond test-time normalization for $18\%$ relative improvement. In concurrent work, Schneider et al. (2020) report $49.3\%$ error for test-time normalization, for which tent still gives $14\%$ relative improvement.

## 4.2 SOURCE-FREE DOMAIN ADAPTATION

We benchmark digit adaptation (Ganin & Lempitsky, 2015; Tzeng et al., 2015; 2017; Shu et al., 2018) for shifts from SVHN to MNIST/MNIST-M/USPS. Recall that unsupervised domain adaptation makes use the labeled source data and unlabeled target data, while our fully test-time adaptation setting denies use of source data. Adaptation is offline for fair comparison with offline baselines.

**Tent adapts to target without source.** Table 3 reports the target errors for domain adaptation and fully test-time adaptation methods. Test-time normalization (BN) marginally improves, while adversarial domain adaptation (RG) and self-supervised domain adaptation (UDA-SS) improve more by joint training on source and target. Tent always has lower error than the source model and BN, and it achieves the lowest error in 2/3 cases, even in just one epoch and without use of source data.

While encouraging for fully test-time adaptation, unsupervised domain adaptation remains necessary for the highest accuracy and harder shifts. For SVHN-to-MNIST, DIRT-T (Shu et al., 2018) achieves a remarkable $0.6\%$ error [2]. For MNIST-to-SVHN, a difficult shift with source-only error of $71.3\%$, DIRT-T reaches $45.5\%$ and UDA-SS reaches $38.7\%$. Tent fails on this shift and increases error to $79.8\%$. In this case success presently requires joint optimization over source and target.

**Tent needs less computation, but still improves with more.** Tent adapts efficiently on target data alone with just one gradient per point. RG & UDA-SS also use the source data (SVHN train), which is $\sim7\times$ the size of the target data (MNIST test), and optimize for 10 epochs. Tent adapts with $\sim80\times$ less computation. With more updates, tent reaches $8.2\%$ error in 10 epochs and $6.5\%$ in 100 epochs. With online updates, tent reaches $12.5\%$ error in one epoch and $8.4\%$ error in 10 epochs.

**Tent scales to semantic segmentation.** To show scalability to large models and inputs, we evaluate semantic segmentation (pixel-wise classification) on a domain shift from a simulated source to a real target. The source is GTA (Richter et al., 2017), a video game in an urban environment, and the target is Cityscapes (Cordts et al., 2016), an urban autonomous driving dataset. The model is HRNet-W18, a fully convolutional network (Shelhamer et al., 2017) with high-resolution architecture (Wang et al., 2020). The target intersection-over-union scores (higher is better) are source $28.8\%$, BN $31.4\%$, and tent $35.8\%$ with offline optimization by Adam. For *adaptation to a single image*, tent reaches $36.4\%$ in 10 iterations with episodic optimization. See the appendix for a qualitative example (Appendix B).

---

[2]We exclude DIRT-T from our experiments because of incomparable differences in architecture and model selection. DIRT-T tunes with labeled target data, but we do not. Please refer to Shu et al. (2018) for more detail.

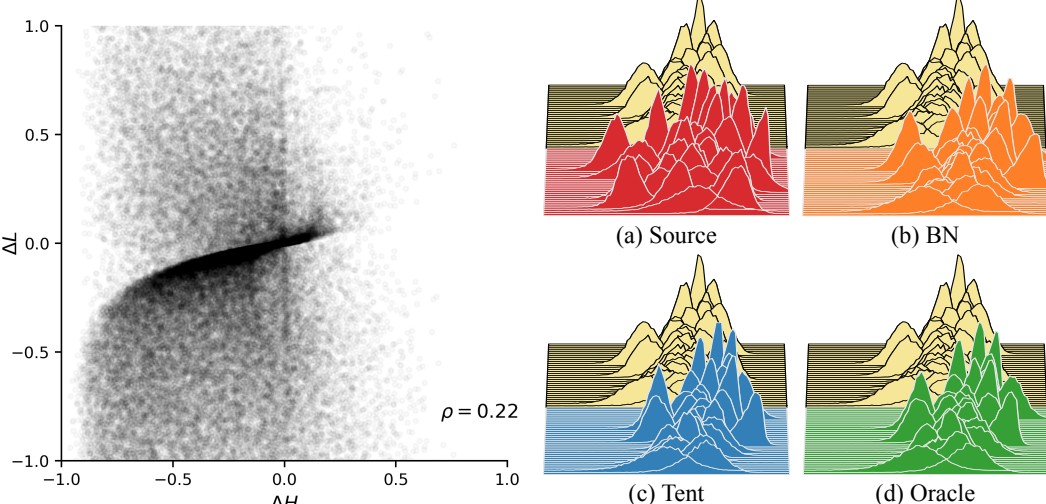

Figure 6: Tent reduces the entropy and loss. We plot changes in entropy $\Delta H$ and loss $\Delta L$ for all of CIFAR-100-C. Change in entropy rank-correlates with change in loss: note the dark diagonal and the rank correlation coefficient of $0.22$.

Figure 7: Adapted features on CIFAR-100-C with Gaussian noise (front) and reference features without corruption (back). Corruption shifts features away from the reference, but BN reduces the shifts. Tent instead shifts features more, and closer to an oracle that optimizes on target labels.

**Tent scales to the VisDA-C challenge.** To show adaptation on a more difficult benchmark, we evaluate on the VisDA-C challenge (Peng et al., 2017). The task is object recognition for 12 classes where the source data is synthesized by rendering 3D models and the target data is collected from real scenes. The validation error for our source model (ResNet-50, pretrained on ImageNet) is $56.1\%$, while tent reaches $45.6\%$, and improves to $39.6\%$ by updating all layers except for the final classifier as done by Liang et al. (2020). Although offline source-free adaptation by model adaptation (Li et al., 2020) or SHOT (Liang et al., 2020) can reach lower error with more computation and tuning, tent can adapt online during testing.

## 4.3 ANALYSIS

**Tent reduces entropy and error.** Figure 6 verifies tent does indeed reduce the entropy and the task loss (softmax cross-entropy). We plot changes in entropy and loss on CIFAR-100-C for all 75 corruption type/level combinations. Both axes are normalized by the maximum entropy of a prediction ($\log 100$) and clipped to $\pm 1$. Most points have lower entropy and error after adaptation.

**Tent needs feature modulation.** We ablate the normalization and transformation steps of feature modulation. Not updating normalization increases errors, and can fail to improve over BN and PL. Not updating transformation parameters reduces the method to test-time normalization. Updating only the last layer of the model can improve but then degrades with further optimization. Updating the full model parameters $\theta$ never improves over the unadapted source model.

**Tent generalizes across target data.** Adaptation could be limited to the points used for updates. We check that adaptation generalizes across points by adapting on target train and not target test. Test errors drop: CIFAR-100-C error goes from $37.3\%$ to $34.2\%$ and SVHN-to-MNIST error goes from $8.2\%$ to $6.5\%$. (Train is larger than test; when subsampling to the same size errors differ by <$0.1\%$.) Therefore the adapted modulation is not point specific but general.

**Tent modulation differs from normalization.** Modulation normalizes and transforms features. We examine the combined effect. Figure 7 contrasts adapted features on corrupted data against reference features on uncorrupted data. We plot features from the source model, normalization, tent, and an oracle that optimizes on the target labels. Normalization makes features more like the reference, but tent does not. Instead, tent makes features more like the oracle. This suggests a different and task-specific effect. See the appendix for visualizations of more layers (Appendix C).

**Tent adapts alternative architectures.** Tent is architecture agnostic in principle. To gauge its generality in practice, we evaluate new architectures based on self-attention (SAN) (Zhao et al., 2020) and equilibrium solving (MDEQ) (Bai et al., 2020) for corruption robustness on CIFAR-100-C. Table 4 shows that tent reduces error with the same settings as convolutional residual networks.

Table 4: Tent adapts alternative architectures on CIFAR-100-C without tuning. Results are error (%).

| SAN-10 (pair) | | | SAN-10 (patch) | | | MDEQ (large) | | |
|---|---|---|---|---|---|---|---|---|
| Source | BN | Tent | Source | BN | Tent | Source | BN | Tent |
| 55.3 | 39.7 | **36.7** | 48.0 | 31.8 | **29.2** | 53.3 | 44.9 | **41.7** |

## 5 RELATED WORK

We relate tent to existing adaptation, entropy minimization, and feature modulation methods.

**Train-Time Adaptation** Domain adaptation jointly optimizes on source and target by cross-domain losses $L(x^s, x^t)$ to mitigate shift. These losses optimize feature alignment (Gretton et al., 2009; Sun et al., 2017), adversarial invariance (Ganin & Lempitsky, 2015; Tzeng et al., 2017), or shared proxy tasks (Sun et al., 2019a). Transduction (Gammerman et al., 1998; Joachims, 1999; Zhou et al., 2004) jointly optimizes on train and test to better fit specific test instances. While effective in their settings, neither applies when joint use of source/train and target/test is denied. Tent adapts on target alone.

Recent "source-free" methods (Li et al., 2020; Kundu et al., 2020; Liang et al., 2020) also adapt without source data. Li et al. (2020); Kundu et al. (2020) rely on generative modeling and optimize multiple models with multiple losses. Kundu et al. (2020); Liang et al. (2020) also alter training. Tent does not need generative modeling, nor does it alter training, and so it can deployed more generally to adapt online with much more computational efficiency. SHOT (Liang et al., 2020) adapts by information maximization (entropy minimization and diversity regularization), but differs in its other losses and its parameterization. These source-free methods optimize offline with multiple losses for multiple epochs, which requires more tuning and computation than tent, but may achieve more accuracy with more computation. Tent optimizes online with just one loss and an efficient parameterization of modulation to emphasize fully test-time adaptation during inference. We encourage examination of each of these works on the frontier of adaptation without source data.

Chidlovskii et al. (2016) are the first to motivate adaptation without source data for legal, commercial, or technical concerns. They adapt predictions by applying denoising auto-encoders while we adapt models by entropy minimization. We share their motivations, but the methods and experiments differ.

**Test-Time Adaptation** Tent adapts by test-time optimization and normalization to update the model. Test-time adaptation of predictions, through which harder and uncertain cases are adjusted based on easier and certain cases (Jain & Learned-Miller, 2011), provides inspiration for certainty-based model adaptation schemes like our own.

Test-time training (TTT) (Sun et al., 2019b) also optimizes during testing, but differs in its loss and must alter training. TTT relies on a proxy task, such as recognizing rotations of an image, and so its loss depends on the choice of proxy. (Indeed, its authors caution that the proxy must be "both well-defined and non-trivial in the new domain"). TTT alters training to optimize this proxy loss on source before adapting to target. Tent adapts without proxy tasks and without altering training.

Normalizing feature statistics is common for domain adaptation (Gretton et al., 2009; Sun et al., 2017). For batch normalization Li et al. (2017); Carlucci et al. (2017) separate source and target statistics during training. Schneider et al. (2020); Nado et al. (2020) estimate target statistics during testing to improve generalization. Tent builds on test-time normalization to further reduce generalization error.

**Entropy Minimization** Entropy minimization is a key regularizer for domain adaptation (Carlucci et al., 2017; Shu et al., 2018; Saito et al., 2019; Roy et al., 2019), semi-supervised learning (Grandvalet & Bengio, 2005; Lee, 2013; Berthelot et al., 2019), and few-shot learning (Dhillon et al., 2020). Regularizing entropy penalizes decisions at high densities in the data distribution to improve accuracy for distinct classes (Grandvalet & Bengio, 2005). These methods regularize entropy during training in concert with other supervised and unsupervised losses on additional data. Tent is the first to minimize

entropy during testing, for adaptation to dataset shifts, without other losses or data. Entropic losses are common; our contribution is to exhibit entropy *as the sole loss* for fully test-time adaptation.

**Feature Modulation** Modulation makes a model vary with its input. We optimize modulations that are simpler than the full model for stable and efficient adaptation. We modulate channel-wise affine transformations, for their effectiveness in tandem with normalization (Ioffe & Szegedy, 2015; Wu & He, 2018), and for their flexibility in conditioning for different tasks (Perez et al., 2018). These normalization and conditioning methods optimize the modulation during training by a supervised loss, but keep it fixed during testing. We optimize the modulation during testing by an unsupervised loss, so that it can adapt to different target data.

## 6 DISCUSSION

Tent reduces generalization error on shifted data by test-time entropy minimization. In minimizing entropy, the model adapts itself to feedback from its own predictions. This is truly self-supervised self-improvement. Self-supervision of this sort is totally defined by the supervised task, unlike proxy tasks designed to extract more supervision from the data, and yet it remarkably still reduces error. Nevertheless, errors due to corruption and other shifts remain, and therefore more adaptation is needed. Next steps should pursue test-time adaptation on more and harder types of shift, over more general parameters, and by more effective and efficient losses.

**Shifts** Tent reduces error for a variety of shifts including image corruptions, simple changes in appearance for digits, and simulation-to-real discrepancies. These shifts are popular as standardized benchmarks, but other real-world shifts exist. For instance, the CIFAR 10.1 and ImageNetV2 test sets (Recht et al., 2018; 2019), made by reproducing the dataset collection procedures, entail natural but unknown shifts. Although error is higher on both sets, indicating the presence of shift, tent does not improve generalization. Adversarial shifts (Szegedy et al., 2014) also threaten real-world usage, and attackers keep adapting to defenses. While adversarial training (Madry et al., 2018) makes a difference, test-time adaptation could help counter such test-time attacks.

**Parameters** Tent modulates the model by normalization and transformation, but much of the model stays fixed. Test-time adaptation could update more of the model, but the issue is to identify parameters that are both expressive and reliable, and this may interact with the choice of loss. TTT adapts multiple layers of features shared by supervised and self-supervised models and SHOT adapts all but the last layer(s) of the model. These choices depend on the model architecture, the loss, and tuning. For tent modulation is reliable, but the larger shift on VisDA is better addressed by the SHOT parameterization. Jointly adapting the input could be a more general alternative. If a model can adapt itself on target, then perhaps its input gradients might optimize spatial transformations or image translations to reduce shift without source data.

**Losses** Tent minimizes entropy. For more adaptation, is there an effective loss for general but episodic test-time optimization? Entropy is general across tasks but limited in scope. It needs batches for optimization, and cannot update episodically on one point at a time. TTT can do so, but only with the right proxy task. For less computation, is there an efficient loss for more local optimization? Tent and TTT both require full (re-)computation of the model for updates because they depend on its predictions. If the loss were instead defined on the representation, then updates would require less forward and backward computation. Returning to entropy specifically, this loss may interact with calibration (Guo et al., 2017), as better uncertainty estimation could drive better adaptation.

We hope that the fully test-time adaptation setting can promote new methods for equipping a model to adapt itself, just as tent yields a new model with every update.

## ACKNOWLEDGMENTS

We thank Eric Tzeng for discussions on domain adaptation, Bill Freeman for comments on the experiments, Yu Sun for consultations on test-time training, and Kelsey Allen for feedback on the exposition. We thank the anonymous reviewers of ICLR 2021 for their feedback, which certainly improved the latest adaptation of the paper.

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

APPENDIX

This supplement summarizes the image corruptions used in our experiments, highlights a qualitative example of instance-wise adaptation for semantic segmentation, and visualizes feature shifts across more layers.

# A  ROBUSTNESS TO CORRUPTIONS

In Section 4.1 we evaluate methods on a common image corruptions benchmark. Table 2 reports errors on the most severe level of corruption, level 5, and Figure 5 reports errors for each corruption type averaged across each of the levels 1–5. We summarize these corruptions types by example in Figure 8.

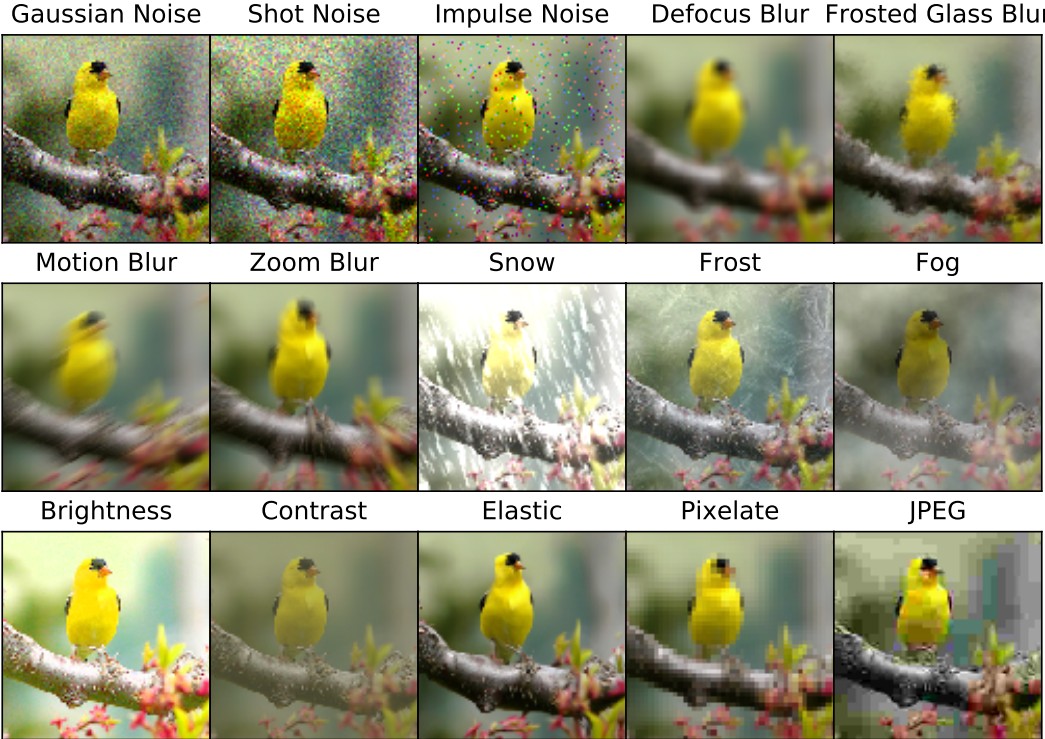

Figure 8: Examples of each corruption type in the image corruptions benchmark. While synthetic, this set of corruptions aims to represent natural factors of variation like noise, blur, weather, and digital imaging effects. This figure is reproduced from Hendrycks & Dietterich (2019).

# B  SOURCE-FREE ADAPTATION FOR SEMANTIC SEGMENTATION

Figure 9 shows a qualitative result on source-free adaptation for semantic segmentation (pixel-wise classification) with simulation-to-real (sim-to-real) shift.

For this sim-to-real condition, the source data is simulated while the target data is real. Our source data is GTA Richter et al. (2017), a visually-sophisticated video game set in an urban environment, and our target data is Cityscapes Cordts et al. (2016), an urban autonomous driving dataset. The supervised model is HRnet-W18, a fully convolutional network Shelhamer et al. (2017) in the high-resolution network family Wang et al. (2020). For this qualitative example, we run tent on a single image for multiple iterations, because an image is in effect a batch of pixels. This demonstrates adaptation to a target *instance*, without any further access to the target *domain* through usage of multiple images from the target distribution.

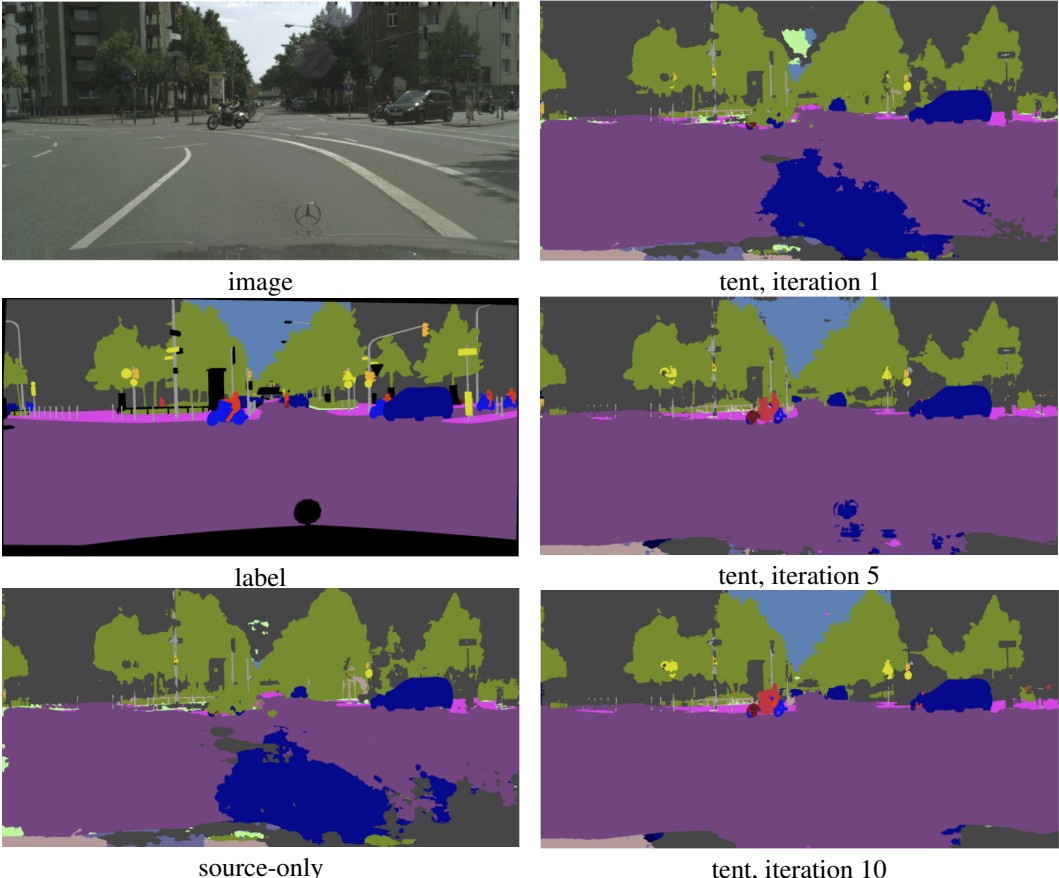

image

tent, iteration 1

label

tent, iteration 5

source-only

tent, iteration 10

Figure 9: Adaptation for semantic segmentation with simulation-to-real shift from GTA Richter et al. (2017) to Cityscapes Cordts et al. (2016). Tent only uses the target data, and optimizes over a single image as a dataset of pixel-wise predictions. This episodic optimization in effect fits a custom model to each image of the target domain. In only 10 iterations our method suppresses noise (see the completion of the street segment, in purple) and recovers missing classes (see the motorcycle and rider, center).

## C  FEATURE SHIFTS ACROSS LAYERS AND METHODS

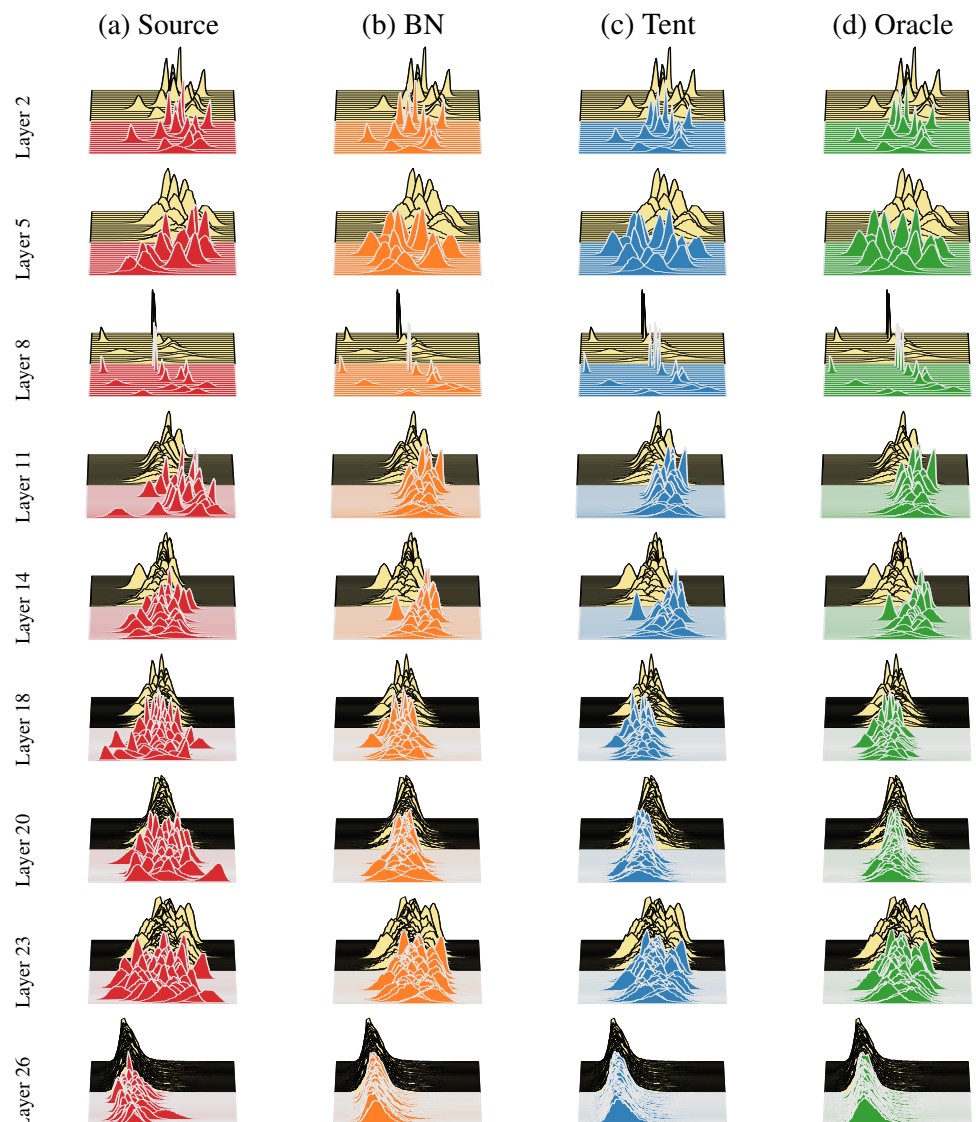

Figure 10: Adapted features on CIFAR-100-C with Gaussian noise (front) and reference features without corruption (back). Corruption shifts the source features from the reference. BN shifts the features back to be more like the reference. Tent shifts features to be less like the reference, and more like an oracle that optimizes on target labels.

