# OpenReview forum: "Tent: Fully Test-Time Adaptation by Entropy Minimization"
_ICLR.cc/2021/Conference — ICLR 2021 Spotlight_

### Official Review · AnonReviewer2 · 2020-10-25
**Interesting and useful setting.**

**Rating:** 8
**Confidence:** 3

**Review:**

Summary: This paper tackles an interesting problem setting — fully test-time adaptation with only target data. The proposed method is to minimize the test-time entropy, and the loss is used to update the feature modulation layer only. The proposed method compares favorably with the state of the arts, on the ImageNet-C benchmark and unsupervised domain adaptation tasks.

Strength:
+ The problem setting is interesting and meaningful
+ The proposed method is simple, efficient, network-agnostic, and generally applicable to many tasks
+ The proposed method achieves competitive performance on the ImageNet-C benchmark and unsupervised domain adaptation tasks

Weakness:
- When comparing with UDA methods (i.e., RG and UDA-SS), I am not sure if the current setting is fair or not. I think they can use the target test set instead of the target training set during training? In this case, it should be a fair comparison with the offline adaptation.
- I wonder if the improvement comes from modulating the features to the target domain distribution, or because the network is optimized specifically on the test data. Run online/offline adaptation on the target training set, and then directly apply this model on the target test set without optimization might be interesting.
- Figure 6 is a bit not intuitive to me. Are the authors trying to convey that “entropy reduction is correlated with classification loss reduction”? Probably a better visualization is needed.
- In Section 4.3 “Tent needs feature modulation”, I wonder where are the exact numbers that I can refer to.
- What is the message the authors try to convey in Figure 7? What does “BN brings them back” mean? I think this study is interesting and important, more discussions and insights are appreciated.

Other comments:
* I wonder when this method will fail. I guess if the target test set is very small, the proposed method might not improve over the source model too much?
* Neural networks sometimes tend to output overconfident (but wrong) predictions even when given out-of-distribution data as input. In this case, I wonder if test entropy is still a good supervision signal to use.

---- Post-rebuttal comments----

Thanks for the response. The rebuttal addresses all my concerns. I am willing to increase my score to 8 and recommend acceptance.

---

> ### Author Response · Authors · 2020-11-19
> **Rebuttal (1/2): adapting to target train vs. test, test entropy and overconfidence**
>
> Thank you for the thoughtful and detailed review! We reply point-by-point here, to begin the discussion, and will post a revised paper tomorrow.
>
> "comparing with UDA methods (i.e., RG and UDA-SS)", "can use the target test set instead of the target training set"
>
> Thank you for the attention to detail. As indicated by Tables 2 & 3, the UDA methods make use of target train while the fully test-time adaptation methods make use of target test. This is by design, as target train may not be available for test-time adaptation, or it may be inefficient for online adaptation.
>
> That said, for rebuttal we control for this difference by experimenting with adapting tent to target train then directly evaluating it on target test without further evaluation. (This is more practical than re-training the UDA methods on target test instead, as they require more computation.) Please see the next point for the results.
>
> "if the improvement comes from modulating the features to the target domain distribution, or because the network is optimized specifically on the test data", "Run online/offline adaptation on the target training set"
>
> This is an intriguing question! For rebuttal we experiment with this by adapting tent online to target train for corruption on CIFAR-100 and domain adaptation on SVHN-to-MNIST. Results improve: CIFAR-100-C error is reduced from 37.3% to 34.2% and SVHN-to-MNIST error is reduced from 8.2% to 6.5%. In this case adaptation is on separate data, but it is also on more data. When target train is subsampled to the size of target test the difference in errors is <0.1%.
>
> As a further check, we experiment with online adaptation without repeating the forward pass (Sec. 3.3) to update predictions. In effect adaptation lags one batch behind, because the predictions are made before the updates. On IN-C, the difference in errors for online tent with and without the repeated forward is negligible at ~0.1%.
>
> In summary, tent does not need to be optimized to the specific test data. Its adapted statistics and transformation parameters generalize across different test samples. We will include these new results and discuss this point in Sec. 4.3 of the revision.
>
> "networks sometimes tend to output overconfident (but wrong) predictions even when given out-of-distribution data", "I wonder if test entropy is still a good supervision signal"
>
> This does indeed happen, in both the corruption and domain adaptation settings. While Figure 2 shows that entropy generally increases on out-of-distribution corruptions, there are particular images with low-entropy and high-loss predictions. Adaptation with tent can nevertheless improve the model predictions on average in spite of these cases as shown by our results (Tables 2 & 3, Figure 5).

---

> ### Author Response · Authors · 2020-11-19
> **Rebuttal (2/2): failure cases, figure clarifications**
>
> "I wonder when this method will fail"
>
> 1. There is no improvement for in-distribution data. Note the equal error on the original ImageNet test set (Figure 5, "source"). Note this is also true of batch norm and TTT, so adaptation within in-distribution data is a pursuit for future work on fully test-time adaptation.
> 2. If there are too many parameters for adaptation, test entropy leads to degenerate solutions. See Sec. 4.3 "Tent needs feature modulation". This is why we parameterize our adaptation through feature modulation (Sec. 3.2).
> 3. If there is too little data for each update, in having a small batch size (32 images or less), the error does not improves. Small batch sizes harm normalization by noisy statistics and harm optimization by trivial solutions. By trivial solutions, we mean that fewer points mean fewer constraints on the shared modulation, so that the modulation need not generalize across target points.
> 4. If the source data is too simple w.r.t. target then tent can fail. For rebuttal, we experiment with MNIST-to-SVHN (the opposite direction of adaptation in the paper) and tent fails to improve error. In fact error increases by 10% relative. The source model has 71.3% error, tent has 79.8%, and the oracle target model has only 4.3%. This is a difficult condition.
> We will include these limitations in the revision for a more comprehensive understanding of tent.
>
> "Figure 6 is a bit not intuitive to me", "entropy reduction is correlated with classification loss reduction?"
>
> Yes. This plot shows the change in entropy vs. the change in loss for every data point in the CIFAR-100-C corruption benchmark. Each point in the scatter plot is a data point in CIFAR-100-C. The relationship between entropy and loss is shown by the rank correlation coefficient and the density of points in the lower-let quadrant near the diagonal. To be clear, it is only a correlation, but this verifies that as entropy is reduced the error is also reduced on average.
>
> "Figure 7?", "What does “BN brings them back” mean?"
>
> This figure (and additional figures in supplement Sec. B) visualize the distribution of feature channels across different adaptation conditions. The message is that the feature distribution for tent is less like batch norm and more like an oracle, where the oracle adapts modulation according to the cross-entropy with the true labels. BN "brings them back" means that the feature distribution for batch norm on corrupted data is closer to the feature distribution for the source model on uncorrupted data. We will expand the paragraph "Tent modulation differs from normalization" in the revision given the expanded limit of 9 pages.
>
> Thank you again for the questions probing how tent succeeds ("modulating the features to the target domain distribution, or because the network is optimized specifically on the test data") and how it can fail ("when this method will fail"). Please let us know if there are further points we can discuss!

---

### Official Review · AnonReviewer4 · 2020-10-27
**Interesting approach, needs some clarifications**

**Rating:** 7
**Confidence:** 5

**Review:**


Presents Test-time Entropy (TENT) minimization, an algorithm for adapting deep models at test time to distributionally shifted data, without requiring access to source training data. At test time, the algorithm updates batch-norm parameters (that control channel-wise normalization and transformation) to minimize predictive entropy over target data. This simple approach is found to lead to state of the art performance on various corruption benchmarks for image classification, and competitive performance on simple DIGITS recognition-based domain adaptation shifts.

Strengths

– The approach appears very simple to implement and seems to work well, particularly on adapting to corruptions

– The paper is well-written, clearly motivated, and very easy to follow. In particular, the source free-assumption is a compelling feature of the method.

– The analysis on rank-correlation b/w change in entropy and loss, and applicability to different architectures, strengthen the claims of the paper

Weaknesses

– While it’s clear that entropy minimization is correlated with correctness, the motivation beyond updating (only) batch-norm parameters to minimize it is unclear to me. Sec 3.2 states that the reason is “stability and efficiency”, but I think a more comprehensive ablation study would validate this choice better.

– I appreciate the fact that the method is benchmarked on multiple tasks – visual corruptions and DA. However the DA experiments/comparisons appear quite cursory. DIGITS is an easy benchmark, and the DA point of comparison (RevGrad) is not competitive with the current SoTA. Without results on other more challenging benchmarks (VisDA, DomainNet, OfficeHome, etc.) I’m not convinced of the usefulness of this method as a DA technique.
To be clear, it would be completely fine if the method *does not* outperform prior work that uses source data or additional computation, but I think it is important to at least benchmark its performance to understand whether it is a viable DA strategy. The qualitative results on semantic segmentation in supplementary appears promising but a quantitative comparison would have been more convincing.

Additional questions / suggestions

– In Fig. 5, how is source performance measured for TENT? Is this performance on source data after applying TENT, with the updated batch-norm parameters? If so, how does TENT achieve identical performance on the source test set?

– It would be interesting to benchmark the performance of TENT for online DA – beyond not requiring source data, being able to also adapt (even reasonably well) online could be very useful.

– A more descriptive caption for Figure 2 would be helpful. Does opacity correspond to the severity of corruption?

– In Fig. 5, it would be good to break down performance of ANT by corruption type as is done for other methods

– Sec 4.2: “Tent needs less computation, but still improves with more”: Does this hold indefinitely? Does performance degrade after a certain number of epochs, or does it remain stable?

Overall comments

Interesting paper on test-time-adaptation that proposes a simple entropy-minimization based objective to update batch norm parameters, and works well on robustness benchmarks. I have concerns around the motivation behind the algorithm’s design, and its viability as a DA method, but would be willing to reevaluate based on the author response.

---- Post-rebuttal comments----

The author rebuttal + revised draft adequately addresses most of my concerns – in particular, the experiments on online adaptation and semantic segmentation are strong, and the additional context on the DA results is helpful. I would still have liked to see DA results on more challenging benchmarks but nevertheless think that the paper proposes an interesting approach and is worth accepting.

---

> ### Author Response · Authors · 2020-11-19
> **Rebuttal (1/2): parameterization and optimization + domain adaptation experiments**
>
> Thank you for the thoughtful and detailed review! We reply point-by-point here, to begin the discussion, and will post a revised paper tomorrow.
>
> "motivation beyond updating (only) batch-norm parameters to minimize it is unclear to me", "more comprehensive ablation study"
>
> We motivate and explain our parameterization in Sec. 3.2 and ablate it in Sec. 4.3 "Tent needs feature modulation". In short, only updating the normalization statistics and affine transformation parameters works while simple alternatives do not. We experiment with optimizing all model parameters, or optimizing the last layer of the network, and both fail for adapting with tent. Our work identifies test entropy and feature modulation as an effective pair for fully test-time adaptation, and we hope more research will identify additional objectives and parameterizations.
>
> "qualitative results on semantic segmentation in supplementary appears promising but a quantitative comparison would have been more convincing"
>
> Note the quantitative results for offline adaptation with the standard metric of intersection-over-union in Sec. 4.2 "Tent scales to semantic segmentation" with source 28.8%, BN 31.4%, and tent 35.8% (higher is better). For rebuttal, we quantitatively evaluate single-image adaptation, as shown in the supplement, and report 36.4% for tent with 10 iterations. Thank you for suggesting this quantitative result, which demonstrates a case for which tent can adapt to a single target instance (one image) rather than needing to observe an entire domain (many images).
>
> "performance of TENT for online DA"
>
> The results on ImageNet-C are for tent with online optimization (Figure 5 and Sec. 4.1 "Tent reaches a new state-of-the-art of 44.0% error by online adaptation and 42.3% error by offline adaptation"). For rebuttal, we also evaluate SVHN-to-MNIST domain adaptation online (compared to offline in Table 3). At one epoch online/offline error is 12.3% vs. 10.0%, but at 10 epochs errors are closer at 8.4% vs. 8.2%. This online capability is a plus for tent, as it is more efficient and lower-latency than offline optimization.
> Is this what the reviewer is requesting, or is "online DA" rather a different setting where domains shift online as the data is encountered?
>
> "Does performance degrade after a certain number of epochs, or does it remain stable?"
>
> Tent largely remains stable, but this depends. For domain adaptation on SVHN-to-MNIST, error keeps improving at 10 epochs to 8.2% (Table 3) and at 100 epochs to 6.5% (evaluated for rebuttal).
> For corruption on CIFAR-100, error worsens slightly from 37.3% at 1 epoch to 38.0% at 10 epochs.
> This is due to our parameterization with feature modulation. With more free parameters, for instance optimizing only the last layer of the network, error first improves and then degrades. If all parameters are optimized, then optimization fails, and error at only 1 epoch is worse than the unadapted source model.
>
> "DIGITS is an easy benchmark", "the DA point of comparison (RevGrad) is not competitive with the current SoTA"
>
> For the benchmark, we choose DIGITS as an accessible benchmark that is common to many works (Ganin et al., Tzeng et al., Sun et al., Li et al., ...).
> For the baselines, we choose RevGrad as a common adversarial method, and UDA-SS as a self-supervised method for more context w.r.t. self-supervised test-time training (TTT). We include these because (1) their model selection rules do not require labeled target data and (2) we were able to reproduce their results. Together these allow for controlled comparison across our baselines and method. To give more domain adaptation context, we will include the stronger accuracy of the DIRT-T UDA method (pointed out by R1) in the revision.
>
> "whether it is a viable DA strategy"
>
> We claim only the feasibility of target-only adaptation (abstract and Sec. 4.2) on digits and simulation-to-real semantic segmentation. We certainly agree that state-of-the-art UDA methods can be stronger and larger domain adaptation benchmarks can be harder. Tent is not a replacement, but an alternative for settings where UDA does not apply without source data or when more efficiency is necessary, and UDA methods help to measure how well tent adapts.
>
> We hope that our results on digits and semantic segmentation encourage research on fully test-time adaptation to pursue more conditions where there is strong progress by unsupervised domain adaptation methods like VisDA and OfficeHome as suggested in the review.

---

> ### Author Response · Authors · 2020-11-19
> **Rebuttal (2/2): experiment and figure clarifications**
>
> "how is source performance measured for TENT?" "how does TENT achieve identical performance on the source test set?"
>
> Tent is applied to source test in the same manner as the other conditions for ImageNet/ImageNet-C: the batch norm stats and affine parameters are adapted online. In the case of source test this yields substantially the same model. That is, tent in effect converges to the source model on source data.
>
> For rebuttal, we experiment with adaptation to CIFAR-10.1 (Recht et al. arXiv'18), which is a new test set for CIFAR-10 made by reproducing its dataset collection procedure. We evaluate and adapt the source model with tent as we do in the paper, and the error is likewise unchanged. This further reinforces that tent does no harm when the target data is similar to the source data.
>
> Sidenote: the results are not quite identical, as the source model on source test is 23.5 while tent is 23.4, but the results are extremely close.
>
> "more descriptive caption for Figure 2", "Does opacity correspond to the severity of corruption?"
>
> Yes, opacity corresponds with severity, and we will relabel "levels" in the legend to "severity" accordingly. We will likewise expand the caption of Figure 2.
>
> "performance of ANT by corruption type"
>
> For rebuttal, we evaluate adversarial noise training (ANT) across corruption types by running the reference implementation. Tent improves on it for all types except noise (gaussian, shot, and impulse) on which ANT is trained. More precisely, gaussian noise is a special case of ANT and the method is initialized to it for adversarial training. We will update Figure 5 in the revision.
>
> Thank you again for the precise questions on domain adaptation and suggestions for figure captions and content. Please let us know if there are further points we can discuss!

---

### Official Review · AnonReviewer1 · 2020-10-28
**A simple method with broad applicability and good results**

**Rating:** 7
**Confidence:** 4

**Review:**

$Paper$ $Summary$

This paper proposes a method to adapt a pre-trained model to a target domain, without the need to access samples from the source domain - on which the model was originally trained. The idea is to adapt layer normalization parameters at test time, by learning affine transformations. This is applied in tandem with the re-collection of the domain statistics.

$Pros$

- The paper is very well written: it is easy to understand the core idea and its applicability in the context of the broader literature. Figures and Tables are also well designed and placed.

- The method is reasonable and simple, and results are strong. As the Authors claim, it is true that the proposed approach has significantly wider applicability than UDA methods and TTT.

$Cons$

- While interesting, I believe Section 4.2 ('Target-only Domain Adaptation') would require additional experiments to properly assess how competitive the proposed approach is with respect to the state of the art of UDA. For instance, several better performing methods could be included in Table 3 (different papers from CVPR/ICLR/etc 2018-2020 achieve significantly higher results on the SVHN -> MNIST split, basically closing the gap with target models - see for example  "A DIRT-T Approach to Unsupervised Domain Adaptation" [Shu et al. ICLR 2018]). The better performing methods still need more data to train (source + target), so higher numbers for the competitors would not undermine the proposed method, but they would provide the reader with a more realistic perspective.

- Results associated with more challenging splits should be provided; for instance, can the proposed method handle MNIST -> SVHN adaptation? This would also clarify one concern I have, which is <how good the source model should be for the method to be effective>. The MNIST -> SVHN split would help clarifying this point, since MNIST models are severely under-performing on SVHN (typically accuracy is ~30%).

- One important baseline that is missing is "Domain Adaptation in the Absence of Source Data", [Chidlovskii et al. SIGKDD 2016]. Can the authors comment on this? It seems to me that some of the Algorithms proposed in this related work could be applied here.

$Minor$ $points$/$suggestions$

- Is the performance of UDA-SS in Table 3 correct? The error is larger than the Source model. I am also checking at the original paper.

- The paper "On Calibration of Modern Neural Networks" [Guo et al. ICML 2017] shows that deep neural networks are poorly calibrated, as over-confident on samples they are wrong about. Since high-confidence generally implies low-entropy, their results are not aligned with Figure 1, that seems to suggest proper calibration. It would be nice to comment on this in the manuscript.

$Review$ $summary$

I believe this is a strong paper, proposing a simple method with large applicability - hence I recommend acceptance. Still, it presents some weaknesses at this stage: I would be happy to read the Author response on these points and iterate the discussion.

---- Post-rebuttal comments----

The rebuttal and the paper revision address my concerns. I fully recommend acceptance.

---

> ### Author Response · Authors · 2020-11-19
> **Rebuttal (1/2): Related Work**
>
> Thank you for the thoughtful and detailed review! We reply point-by-point here, to begin the discussion, and will post a revised paper tomorrow.
>
> "how competitive the proposed approach is with respect to the state of the art of UDA", "see for example "A DIRT-T Approach to Unsupervised Domain Adaptation" [Shu et al. ICLR 2018"
>
> Thank you for the pointer to a stronger unsupervised domain adaptation (UDA) method.
> We will cite and discuss it in the revision to give further context to our adaptation results.
>
> Our position is that UDA methods serve as context for our work, not competition, since tent makes use of less data and computation. We claim in the abstract and Sec. 4.2 that target-only adaptation is feasible, and can rival UDA, but that does not mean it is better: it addresses a different setting. The UDA results help gauge the performance of fully test-time adaptation.
>
> Therefore it is valuable to know that the state-of-the-art for UDA is better still, to underline the opportunity for more research on fully test-time adaptation to try and close the gap. While we will note the accuracy of DIRT-T in the text of the revision, it does not seem appropriate to compare with it directly in the tables as the experimental conditions are different. DIRT-T's impressive accuracy is in part due to more hyperparameters (three loss coefficients, plus optimization settings) and cross-validation on labeled target data, while our baselines and method do not depend on labeled target data for tuning, and so are not comparable.
>
> "Domain Adaptation in the Absence of Source Data"[Chidlovskii et al. SIGKDD 2016]", "algorithms proposed in this related work could be applied here."
>
> Thank you for this early reference motivating adaptation without source data. We will cite it in the revision: "To the best of our knowledge, Chidlovski et al. '16 are the first to motivate adaptation without source data due to legal, contractual, or technical limitations. They adapt classifier predictions without adapting the classifier itself by applying denoising auto-encoders, in contrast to end-to-end source-free methods that adapt the model itself."
>
> Chidlovskii et al. differ in their method and experimental conditions: they restrict their attention to linear classifiers (on fixed deep features) and do not consider corruptions. Reproducing their method and extending it into an end-to-end method is out of scope for our work, but we confirm that we will cite it and thank you again for the reference.
>
> Thank you again for the two references on unsupervised domain adaptation and source-free adaptation. Please let us know if there are further points we can discuss!

---

> ### Author Response · Authors · 2020-11-19
> **Rebuttal (2/2): Domain Adaptation Experiments**
>
> "how good the source model should be for the method to be effective?", "can the proposed method handle MNIST -> SVHN adaptation"
>
> The error rates of the source models on target vary considerably, but tent is able to improve them. In particular, ImageNet-C and CIFAR-100-C for corruption have more errors than correct predictions at 59.5% and 67.2% error, while two of the digits datasets for domain adaptation (MNIST, USPS) are mostly correct with error rates of only ~18% (see Tables 2 & 3).
>
> However, we caution that the source model predictions can be too poor for improvement. For rebuttal we attempted domain adaptation from MNIST-to-SVHN as requested: tent fails to improve error. In fact error increases by 10% relative. The source model has 71.3% error, tent has 79.8%, and the oracle target model has only 4.3%. This is a difficult condition in which other methods also fail (RevGrad, ADDA, …), although DIRT-T and UDA-SS notably succeed with <40% error.
>
> We will include this failure in the revision, as an instructive case for when unsupervised domain adaptation may be necessary (recall that DIRT-T and UDA-SS jointly use source and target, while tent only uses target). Thank you for suggesting this experiment.
>
> "Is the performance of UDA-SS in Table 3 correct?"
>
> We too were concerned with this result, because while UDA-SS reduces error for corruption (Table 2) it increases error for domain adaptation (Table 3). We identified a bug in the reference code for UDA-SS, which the authors of UDA-SS confirmed after the deadline, and we have now fixed it. The fix (1) makes optimization more stable (2) improves domain adaptation error (3) slightly harms corruption error. The fix in fact improves results over the UDA-SS paper for SVHN-MNIST.
>
> - For corruption (Sec. 4.1) UDA-SS errors is 16.7 on CIFAR-10-C (vs. 15.2) and 47.0 on CIFAR-100-C (vs. 44.0).
> - For DA (Sec. 4.2) UDA-SS errors are 11.1 on MNIST (vs. 39.0) 22.2 on MNIST-M (vs. 44.1) and 18.4 on USPS (vs. 22.8)
>
> These improved errors maintain the pattern of results at the time of submission: tent improves on the UDA baselines for corruptions and improves on fully test-time baselines for digit domain adaptation while sometimes rivaling UDA baselines although tent uses less data.
>
> Thank you again for the precise suggestions on improved domain adaptation experiments. Please let us know if there are further points we can discuss!

---

> > ### Comment · AnonReviewer1 · 2020-11-21
> > **Response to rebuttal**
> >
> > Many thanks for addressing all of my concerns.
> >
> > I am not worried by UDA methods performing better than the proposed approach in some settings, it is reasonable that the proposed method fails to cope with a difficult problem such as MNIST --> SVHN. On the other hand, it is important to include such result in the manuscript, to avoid a misleading interpretation -- I appreciate the Authors doing it.
> >
> > After having read the Author response, the revised parts of the manuscript (Sections 4.2 and 5), and other Reviewers' comments, I confirm my positive feedback on this work: In my opinion, it should be accepted for publication.

---

### Comment · ~Shaohua_Li2 · 2021-01-28
**Good paper, but missing an important reference**

Part of the idea, i.e., doing domain adaptation by minimizing test domain entropy has been proposed in "ADVENT - Adversarial Entropy Minimization for Domain Adaptation in Semantic Segmentation", CVPR 2019. This paper is very relevant but not cited in the paper.

---

> ### Author Response · Authors · 2021-01-29
> **Thank you, ADVENT is related and we will cite it**
>
> Thank you for your attention and for sharing this relevant reference! ADVENT is related to tent by its entropy minimization for adaptation. Note however that it addresses the standard unsupervised domain adaptation (UDA) setting, with joint offline optimization over source and target, and not our fully test-time adaptation (FTTA) setting. We additionally consider adaptation to a single target image at a time, rather than the entire target set, which is made possible by our online or episodic updates.
>
> We will cite ADVENT under entropy minimization in our related work. We will also mention its accuracy alongside our GTA-Cityscapes results as context for UDA vs. FTTA adaptation. ADVENT can achieve a higher accuracy with more data, so this can motivate more research on FTTA to improve its accuracy.
>
> (As an aside, ADVENT's discriminator for the alignment of uncertainties across source and target is an interesting further usage of entropy, but again it does require joint access to source and target.)

---

### Comment · ~Yuntao_Du.1 · 2021-02-13
**Good paper, but missing some related baselines**

The setting of this paper is a realistic problem where only a pre-trained model and target data are given. The setting is firstly proposed in paper "Besides the paper "Domain Adaptation in the Absence of Source Data", and some recent works have also been proposed with the setting named "source-free domain adaptation". I  think these works may be the baselines for this paper.

[1]Sahoo, Roshni, Divya Shanmugam and John Guttag. “Unsupervised Domain Adaptation in the Absence of Source Data.”  KDD 2016

[2]Liang, Jian, D. Hu and Jiashi Feng. “Do We Really Need to Access the Source Data? Source Hypothesis Transfer for Unsupervised Domain Adaptation.” ICML 2020

[3]Li, Rui, Qianfen Jiao, Wenming Cao, H. Wong and Si Wu. “Model Adaptation: Unsupervised Domain Adaptation Without Source Data.” 2020 IEEE/CVF Conference on Computer Vision and Pattern Recognition (CVPR) (2020): 9638-9647.

---

> ### Author Response · Authors · 2021-03-18
> **Thank you, source-free adaptation is closely related and we have expanded our discussion of it**
>
> Thank you for pointing out recent and concurrent work on source-free adaptation. These are certainly related, as fully test-time adaptation and source-free adaptation both adapt on target data alone. That said, fully test-time adaptation places additional emphasis on adapting online during testing. Existing source-free methods are offline, and so inference must wait until their optimization completes, while tent adapts and infers at the same time. We have updated our related work and discussion to clarify this point.
>
> In regard to the specific references given:
>
> - [1] is a promising first step for test-time adaptation of the input rather than test-time adaptation of the model. We encourage more work on input adaptation, model adaptation, and ideally joint adaptation over both! Note that [1] and tent are concurrent on arxiv.
>
> - [2] addresses the same source-free setting as [3] and Kundu et al. (cited in our paper). As it includes entropy minimization as one of its losses, we now discuss it more in our related work, and  highlight it alongside our result on VisDA-C.
>
> - [3] was already cited, but is likewise now given further coverage in our related work, and its better accuracy is mentioned alongside our result on VisDA-C.

---

### Decision · Program_Chairs · 2021-01-07
**Final Decision**

**Decision:**

Accept (Spotlight)

**Comment:**

The paper is proposing a test time adaptation method without modifying the training. The proposed idea is simple and effective, adapting the normalization layers using the entropy of the model predictions as a loss function. The paper presents an extensive empirical study. Paper received unanimously accept scores. It also has potential to be impactful as it is easy to apply without any strong assumption/requirement. A clear accept!